# For Heart Rate Assessments from Drone Footage in Disaster Scenarios

**DOI:** 10.3390/bioengineering10030336

**Published:** 2023-03-07

**Authors:** Lucas Mösch, Isabelle Barz, Anna Müller, Carina B. Pereira, Dieter Moormann, Michael Czaplik, Andreas Follmann

**Affiliations:** 1Department of Anesthesiology, Faculty of Medicine, RWTH Aachen University, 52074 Aachen, Germany; 2Institute of Flight System Dynamics, RWTH Aachen University, 52062 Aachen, Germany

**Keywords:** contactless, mass-casualty incident, drones, RGB video, signal, extraction, triage

## Abstract

The ability to use drones to obtain important vital signs could be very valuable for emergency personnel during mass-casualty incidents. The rapid and robust remote assessment of heart rates could serve as a life-saving decision aid for first-responders. With the flight sensor data of a specialized drone, a pipeline was developed to achieve a robust, non-contact assessment of heart rates through remote photoplethysmography (rPPG). This robust assessment was achieved through adaptive face-aware exposure and comprehensive de-noising of a large number of predicted noise sources. In addition, we performed a proof-of-concept study that involved 18 stationary subjects with clean skin and 36 recordings of their vital signs, using the developed pipeline in outdoor conditions. In this study, we could achieve a single-value heart-rate assessment with an overall root-mean-squared error of 14.3 beats-per-minute, demonstrating the basic feasibility of our approach. However, further research is needed to verify the applicability of our approach in actual disaster situations, where remote photoplethysmography readings could be impacted by other factors, such as blood, dirt, and body positioning.

## 1. Introduction

The possibilities for using unmanned aerial systems (UAS) or drones in health emergencies have increased [1]. UASs can cover short-to-medium distances in a relatively short time and access locations that are difficult for rescuers to reach [2,3] while avoiding environmental hazards [4,5]. During a mass-casualty incident (MCI), the situation on-site, the number and location of injured people, and the possible evacuation routes have been evaluated successfully using drone-camera footage [6]. The systems used ranged from conventional cameras, with live feeds to aid in decision-making [7], to thermal cameras, used to locate missing people [8], and 3D-camera setups for reconstructing a disaster scene [9].

The current procedure during MCIs is for responders to physically evaluate injured persons to assess their condition.This assessment is part of the triage process, where the severity of their injuries categorizes the injured. However, depending on the accessibility of the scene, it can take from a few minutes up to several hours for emergency personnel to locate and categorize injured people. In order to assess their condition, basic vital signs, such as heart rate (HR) and respiratory rates (RR), are measured. The gold standard for assessing HR is palpation, which requires direct contact with the victim.

The contactless assessment of vital parameters via drones could save crucial time, as obstacles do not impair drones. Furthermore, UAVs have the potential for faster and more efficient assessments of individual health states in remote and hard-to-reach locations. Our presented research aimed to investigate the technical possibilities, the capabilities, and the limitations of drone-based heart-rate assessments in order to provide a basis for future drone-assisted triage.

Several research publications have shown that camera-based HR measurements are feasible [10,11]. The developed algorithms were designed for different scenarios, such as monitoring pre-term babies in neonatal intensive care units [12], monitoring patients in intermediate care units [13], detecting infections (COVID-19) [14], and monitoring drivers [15].

Image photoplethysmography (iPPG), also called remote photoplethysmography (rPPG) or distance photoplethysmography, is a widely used method for the camera-based assessment of HRs [16,17]. Similar to contact-PPG, rPPG measures variations in the intensity of the light reflected by the skin [18]. The environmental light and the video camera substitute the light source and photo-detector, respectively. The rPPG signal is then computed from consecutive video frames, primarily using facial regions-of-interest (ROIs), such as the forehead and cheeks [19,20]. The main challenge in estimating rPPG signals with camera systems is the extremely low signal strength of the pulse-related intensity changes, as compared to those caused by environmental lighting or system motion. In order to obtain a reliable HR, several different methods were combined via multi-stage algorithmic pipelines. The current research focused on three main methods:(i)Selection and temporal tracking of relevant facial regions;(ii)Modeling of the recorded signals for PPG signal extraction;(iii)De-noising of rPPG signals.

Regarding the selection of facial regions and temporal tracking (i), Kumar et al. proposed a method using a Haar-cascade-based facial detection [21] and a Kanade–Lucas–Tomasi (KLT) feature tracker [22,23] to identify and track candidates with suitable rPPG regions. Those regions were then further weighted to reduce the impact of regions that could contain noisy signals. For modeling the pulse-related color changes (ii), there are two standard models, namely CHROM (Chrominance), proposed by De Haan et al. [24], and POS (Plane Orthogonal to Skin), proposed by Wang et al. [25]. Both models use a pre-defined skin-color vector with an adaptive whitening to separate the recorded RGB signals into an rPPG signal component and an orthogonal noise component. To further improve their approach, Wang et al. added a de-noising step (iii) to their rPPG model, and called their result discriminative signatures [26]. For de-noising, the authors used generalized linear modeling (GLM) to remove the first-order components of known noise signals from the recorded PPG signals. As the only known noise source for these signals, the motion recorded by the facial landmark tracking was used.

Although current approaches have performed considerably well, they also required a highly controlled environment with minimal interference to guarantee reliability. They could only extract reliable vital parameters when a person moved as little as possible. Furthermore, they were prone to changes in lighting conditions and variations in capturing angle. The utilization of these methods in outdoor scenarios poses the challenge of mitigating the effects of uncontrollable ambient lighting on the obtained data. Additionally, when utilizing drone-mounted systems, an additional difficulty is introduced in order to compensate for the system’s increased motion and the increased distance between the system and the subjects being monitored. These challenges had to be addressed to ensure the obtained data’s accuracy and reliability.

This paper proposed a novel algorithmic pipeline specifically developed and optimized for estimating HRs from videos captured by a UAS. The primary objective of this research was to evaluate the potential of UAS-based rPPG as a first step in the assessment of the physiological status of victims in MCIs by leveraging the capability of UAS to reach the disaster location in a shorter time than first-responders.As a secondary objective, we investigated the selection of appropriate de-noising sources to compensate for the system movement and to study the effects of environmental lighting on HR assessments from rPPG signals. Additionally, we provided a full implementation of this approach, along with a proof-of-concept study that could improve patient outcomes and incident scene management.

## 2. Materials and Methods

### 2.1. Imaging Pipeline Overview

The following sections present the estimation of vital parameters via drone videos. The concept consisted of five steps:(i)**Data acquisition and image pre-processing:** Adjustments made to the system’s parameters before and during the measurements;(ii)**ROI selection and tracking:** Selection of beneficial facial regions to track throughout the recording to extract rPPG signals;(iii)**rPPG signal extraction:** Combining the measured raw data from which the rPPG signals would be extracted;(iv)**De-noising and post-processing:** Refinement of the extracted rPPG signals to obtain a predominantly sparse signal for assessing HR;(v)**Heart-rate estimation:** Final step of calculating the HR from the extracted rPPG signals.

### 2.2. Data Acquisition and Image Pre-Processing

Data acquisition and pre-processing were the crucial parts of the pipeline, as any deficiencies would directly impact all subsequent steps. Our goal was to obtain the most stable possible recording of an individual’s head. The recording was conducted on the raw data format because video encoding with common H.264 (advanced video coding) or H.265 (high-efficiency video coding) encoders could impair the subsequent HR assessments [27].

#### 2.2.1. Face-Aware Adaptive Exposure Time Adjustment

The intensity of the rPPG signals was proportional to the skin-reflected light [25]. The greater this intensity, the higher their ratio, as compared to the ambient and sensor noise (signal-to-noise ratio, SNR). The amount of light captured could be controlled through the camera’s exposure time. Integrated exposure time algorithms usually attempt to generate evenly illuminated recordings. However, we experienced that integrated algorithms exposed facial regions either too much or too little. Therefore, we developed a software-driven exposure algorithm that maximized the exposure of facial regions by applying facial detection before the recording.

The images provided by the camera were denoted as $I=[i_{t=0},\ldots ,i_{N}]$, and the corresponding exposure time was $L=[l_{t=0},\ldots ,l_{N}]$. Empirical evidence showed that to improve the SNR of the pulse signal, the 98th percentile of the pixel values within the face region *v* had to be between 75% and 90% of the gray-scale range, i.e., between the pixel values 190 and 230, for an 8-bit image. The goal was to obtain as much exposure as possible but, at the same time, prevent signal clipping. For obtaining the region containing the face-related pixels, pre-existing facial recognition approaches based on Haar cascades [21], HOG + linear SVM [28], fast R-CNN [29], and YOLO [30] were tested. The facial recognition algorithm had to operate on the hardware integrated into the drone, which had very limited computing power. Therefore, simple Haar cascades were used since they enabled sufficient facial recognition at a higher computational speed. Since the relationship between exposure time $e_{n}$ and facial pixel would not be known beforehand, we used the iterative adjustment algorithm (Algorithm 1) after using the cameras initially provided $l_{0}$.
**Algorithm 1** Automatic exposure time adjustment with respect to a person’s face1:**let** $i_{0}$ the initial image frame as gray image2:**let** ${\mathrm{bb}}_{0}$ the initial facial bounding box within $i_{0}$3:**let** $l_{0}$ the initial exposure time4:**for** n=0 **to** 10 **do**5:    $v=percentile(pixel\left({\mathrm{bb}}_{n}\right),98)$6:    **if** 190<vs.<230 **then**7:        **return** $l_{n}$8:    **else**9:        $l_{n+1}\leftarrow \frac{190+230}{2v}\cdot l_{n}$10:        $l_{n+1}\leftarrow image\left(l_{n+1}\right)$11:        ${\mathrm{bb}}_{n+1}\leftarrow haarcascade\left(l_{n+1}\right)$12:    **end if**13:**end for**14:**return** 
$l_{n}$

#### 2.2.2. Active Image Stabilization

During operation, drones have to withstand a variety of weather conditions. Depending on the wind gusts, the position relative to the target could dynamically change. The deviations could vary by several centimeters, thus creating additional movement, vibration, and shake during the recordings. Electromechanical gimbals compensated by maintaining a constant visual axis relative to the target. For this purpose, the gimbals used a motor-driven cardanic suspension that immediately compensated for the smallest movements with the assistance of acceleration sensors and an active control system. With a three-axis gimbal, only perspective distortions occurred during the recordings of the mounted cameras. Since we were restricted to using a two-axis gimbal that could only compensates the pitch and roll axes, the captured images were additionally rotated by a certain degree, which required additional compensation. For this, we used the the view-axis related to the left-facing vector $\vec{h}$ and the down-facing vector $\vec{v}$ in geodetic coordinates provided by the gimbal, as shown in Figure 1.

As a first step, the synchronized pairs of the gimbals and image data were created, as the acquisition of both was usually not synchronous and performed at different frequencies. For this, we used the provided acquisition timestamps $t_{n}^{i}$ of the images and $t_{m}^{g}$ of the gimbal vectors. For each image–timestamp pair $\left(i_{n},t_{n}^{i}\right)$, the corresponding gimbal-vectors $\left(\vec{h_{m}},\vec{v_{m}},t_{m}^{g}\right)$ were determined by minimizing the temporal distance: (1)$$|t_{n}^{i}-t_{m}^{g}|\to min.$$With the corresponding gimbal vectors, the rotation of the optical plane $y_{n}$ of each image could be calculated using the z-component of $\vec{h_{m}},\vec{v_{m}}$ by: (2)$$y_{n}=atan2\left(h_{m}^{z},v_{m}^{z}\right).$$Afterward, each image $i_{n}$ was digitally rotated by $-y_{n}$ using bilinear interpolation to remove the image rotation caused by the gimbal. All images were centered within a zero-padded quadratic frame of a squared-image width to maintain a rectangular image shape. Therefore, a new set of quadratic, rotated images $I_{\mathrm{rot}}=\left[i_{\mathrm{rot},t=0},\ldots ,i_{\mathrm{rot},N}\right]$ of equal size were created, as demonstrated in Figure 2.

### 2.3. ROI Selection and Tracking

The next step of our approach was to process the rotation-free images further by identifying and tracking the facial ROIs. Figure 3 displays the four major steps used for this purpose.

First, a facial bounding box within each image of $I_{\mathrm{rot}}$ was calculated to extract the facial area (Figure 3, Step 2). The entirety of the facial areas resulted in a time series of the facial size and position $B=[{\mathrm{bb}}_{t=0},\ldots ,{\mathrm{bb}}_{N}]$. Here, the same Haar-cascade-based classifier used for the adaptive exposure time adjustment was used for the calculation. However, due to the movements of the UAS, only some images could contain a face entirely; thus, face detection inevitably failed when this was not the case. As a result, the time series could be discontinuous, creating several time-frames of consecutive facial areas $B_{t=[a,b]},B_{t=[a^{\prime },b^{\prime }]},\ldots$ for the same recording. Therefore, all time-frames of facial areas that were at least five seconds long were used for further processing. In cases where no such time-frame existed, the assessment of the HR was declared as failed. Within the facial area of each time-frame, the facial ROIs required for the rPPG extraction were determined using a 3D-mesh representation of the face (Figure 3, Step 3). For this purpose, the face mesh function of Google’s Media Pipe framework was used [31,32]. The provided function combined a neural network for facial feature extraction with a feature-based 3D transformation to generate 468 3D landmarks from 2D facial images in order to create a face mesh. In the end, 30 sub-regions covering the forehead and cheek regions were selected within this face mesh. They represented the ROIs used for extracting rPPG signals. Finally, the raw rPPG signal was determined for each ROI by averaging the pixel color values within the region throughout the video frames of the time series.

### 2.4. De-Noising and Signal Post-Processing

The raw signals extracted from the different ROIs were usually too noisy to be used as rPPG signals. In general, the noise was related to movements caused by wind and wind changes and the inherent vibrations of the drone, for which the gimbal could not fully compensate. It was also related to the perspective distortions and changing lighting conditions. Therefore, a generalized linear-model-based de-noising was developed to reduce the noise caused by the UAS and the environmental changes in lighting. This de-noising consisted of modeling the raw signal of each color channel and each ROI $y_{R},y_{G},y_{B}$ as the composition of a linear mixture of known noise source signals X through the mixing matrix m. The blood flow correlated to the rPPG signal p and the remaining unknown noise component e, as expressed by: (3)$$\begin{array}{ccc} y_{1} & = & m_{1}x_{1,1}+m_{2}x_{2,1}+\ldots +p_{1}+e_{1}\\ y_{2} & = & m_{1}x_{1,2}+m_{2}x_{2,2}+\ldots +p_{2}+e_{2}\\ y_{3} & = & m_{1}x_{1,3}+m_{2}x_{2,3}+\ldots +p_{3}+e_{3}\\ \; & \vdots & \vdots \\ y_{n} & = & m_{1}x_{1,n}+m_{2}x_{2,n}+\ldots +p_{n}+e_{n}, \end{array}$$
where $y_{1}\ldots y_{n}$ denotes the temporal components of an ROI’s color component and $m_{1}x_{1},m_{2}x_{2},$*…* denotes the weighted components known source of noise. Assuming an equally known source of noise for each color channel, this resulted in: (4)$$\begin{array}{ccc} y_{R} & = & X\cdot m_{R}+p_{R}+e_{R}\\ y_{G} & = & X\cdot m_{G}+p_{G}+e_{G}\\ y_{B} & = & X\cdot m_{B}+p_{B}+e_{B}. \end{array}$$To determine the noise contribution components of m, the former equation could be re-written as: (5)Y=X·M+P+E.To solve for M, a standard least-squares approach for over-determined systems was chosen. With this contribution matrix, the raw signal Y could be de-noised by subtracting the noise contributions: (6)$$\hat{Y}=Y-X\cdot M=P+E,$$
where $\hat{Y}$ represents the de-noised signals. Since the de-noising performance of this approach depended solely on the selection of appropriate known noise sources, their selection was crucial and had to satisfy two criteria. On the one hand, the noise sources had to correlate with the actual noise influence within the raw signal. On the other hand, they could not correlate with the blood-flow-related changes, as this removed them during the de-noising process. In addition, too many noise sources could ruin the de-noising process, as the accumulated noise could also lead to the removal of blood-flow-related signal components.

For our approach, 15 sources of noise were used, as listed in Table 1. To compensate for noise components correlated with the unwanted movement of the UAS, data provided by the flight systems sensors were used, namely from the GPS and the accelerometer. They were provided and pre-processed identically to the gimbal data by matching them to the corresponding timestamps. Therefore, matching time series were created from 6 noise source signals that included (1) **height above ground**, (2) **longitude**, (3) **latitude**, and (4) the (u,v,w) values of the **velocity vector** in world coordinates. In addition, the time series of the (x,y,z) values of the two gimbal vectors (5) $\vec{h}$ and (6) $\vec{v}$ were included as noise source signals, as well.

For compensation of the movement correlated with the tracking of facial regions, the (x,y) values of the (7) **landmarks** forming the facial ROI and their (8) **center-of-mass** were utilized. The number of landmarks used varied between four and six, depending on the ROI. Moreover, the (9) **number of pixels** included in the facial ROI was used as well. Finally, the mean color values of the facial regions not containing blood-related color changes were utilized to account for other noise influences. These were affected by other noise influences, such as changes in lighting, in the same way as the facial ROIs but did not contain blood-related color changes. Therefore, they could be used as direct noise sources that accounted for these influences. For this, the sub-regions covering the (10) **nostrils** and the (11) **eyes** were processed in the same way as the facial ROIs by using the time series of their mean color values. Thus, four additional noise source signals were created for de-noising.

The linear modeling of the de-noising problem removed only linearly correlated components. To consider higher-order influences, we also used the first-order derivatives of the (12) **landmarks**, (13) **center-of-mass**, and (14, 15) **gimbal vectors**. As a result, depending on the number of landmarks creating the sub-regions, 42 to 50 individual signals were created and used for de-noising.

With the matrix M consisting of these known noise signals, every color channel of the 30 sub-ROIs was de-noised independently, creating a new set of de-noised time-frame signals. The last part of this step consisted of generating an rPPG signal for each region and time-frame, i.e., continuous video sequences with a length of at least 5 s. For this purpose, the individual color signals had to be combined to form an rPPG signal. In the literature, several methods existed. The more sophisticated approaches aimed to obtain the blood-flow-related components through color channel mixing. This work applied two approaches: POS [25] and CHROM [24]. Both algorithms attempt to determine the color vector most closely corresponding to the blood-related color changes by estimating the face color in order to project the color signals onto it, using the variance of the power spectrum of the changes. In our approach, both rPPG signals created by POS and CHROM were calculated and used for robust HR assessments.

### 2.5. Heart-Rate Estimation

Most of the current rPPG approaches focus on the continuous monitoring of patients, and consequently, they focus on detecting variations in HR. However, in disaster medicine, the temporal variation of this parameter can be less relevant than the immediate assessment of an average value for patient triage. Therefore, our approach focused on obtaining a single value for HR.

For an HR assessment, it was assumed that an HR fluctuated only slightly over a short period. Hence, equivalent blood-related changes could be measured at any time and in any facial ROI with both color-mixing methods. In Figure 4, we detailed how this was utilized.

First, each de-noised ROI signal was sliced into overlapping sub-signal windows of 45 samples (i.e., 5 s) with a stride of 2 samples, and the frequency spectrum $P_{n}$ of each was calculated using the Fourier transform (Figure 4, (1)). Second, improper sub-signal spectra were removed from this set of spectra. Each spectra had to satisfy two conditions to be considered for HR estimation (Figure 4, (2)). As a first condition, any frequency peak within the target range of 50 bpm (beats-per-minute), up to 180 bpm, had to have an SNR of at least 2.5%. For this, a prominence-based peak search was conducted within the target frequency range, followed by the SNR calculation for each peak *p*: (7)$$SNR=\frac{\sum_{p-\alpha }^{p+\alpha }P\left(p\right)}{\sum P-\sum_{p-\alpha }^{p+\alpha }P\left(p\right)}.$$For the peak width α, a value of 3 bpm was chosen. By adding this criterion, we intended to exclude spectra that were too noisy from the HR assessment. As a second condition, there had to be no global maximum regarding the target range within the low-frequency range between 0 bpm and 50 bpm. This condition prevented the higher harmonics produced by prominent low-frequency noise components from concealing the HR-related frequency within the target range.

Third, all remaining spectra were combined for CHROM and POS, via a median, resulting in two total spectra each. A resulting HR value was then determined for both total spectra by determining the frequencies corresponding to the respective maximums in the range of the target range. The final HR value was then calculated by the mean value of the two HR values determined via POS and CHROM. The mean value was chosen because if the HR values of the two methods differed, it could not be determined which was correct. By averaging, the values correctly recorded by one method could worsen. However, the substantial deviations from the actual value were reduced, making the robust detection of a single value more valuable.

### 2.6. Experimental Evaluation

#### 2.6.1. Flight System and Camera Setup

The drone used for the experimental study was a tilt-wing UAS developed by the Institute of Flight System Dynamics (FSD), RWTH Aachen University, and the flyXdrive GmbH (FXD), in Aachen, Germany. It was based on the “Neo” tilt-wing UAS manufactured by FXD and was provided within the FALKE research project (funded by the Federal Ministry of Education and Research).

Tilt-wing aircraft are characterized by their ability to perform efficiently and fast forward-flight for bridging longer distances as well as hover-flight, allowing vertical takeoff and landing almost anywhere. Depending on the flight mode, an operation time of up to one hour can be achieved, whereby in pure hovering flight, the system would be limited to a shorter operation time. During the fixed-wing flight, a top speed of up to 130 km/h could be achieved. During the research project, the drone, displayed in Figure 5, was further adapted by the consortium to include sensors, and a gimbal was required towards the front of the aircraft for the remote assessment of vital signs. The 2-axis gimbal was integrated into the system to compensate for system motion during video recordings. This could be controlled on the roll-axis, from −90° to +90°, and the pitch-axis, from −15° to +60°, with an actuation speed of 400 °/s.

The UAS was equipped with an Allied Vision Mako G-234C RGB (Allied Vision Technologies GmbH, Stadtroda, Germany) camera utilizing a Sony IMX249 CMOS sensor (Sony Group Corporation, Tokyo, Japan). This was chosen because it had a volume of 28 ccm, weighed 80 g, could capture raw 10-bit Bayer frames at up to 41.2 frames-per-second (fps), and had a special resolution of up to 1936-by-1216 pixels. In addition, it had a typical C-mount that could be used to mount any optics that could enable it to function at various distances.

#### 2.6.2. Data Acquisition

For the recordings, the camera was equipped with a Kowa 50 mm 1" 5MP C-mount lens (Kowa Optimed Deutschland GmbH, Düsseldorf, Germany). Frame capturing was conducted in an 8-bit Bayer format of 1920-by-1080 pixels at a frame rate of 15 fps. For managing and storing the recordings, a NanoPC-T4 (open-source) operating a custom recording software was integrated into the UAS.

#### 2.6.3. Experimental Setup and Study Protocol

To experimentally evaluate our system for HR assessments, a group of 18 subjects was recorded with the UAS during flight under different conditions. The experiments were conducted with the assistance of healthy volunteers. The ethics committee of the medical faculty of the RWTH Aachen University in Germany approved the conduct of the study. Informed consent was obtained from all subjects involved in the study. The group of volunteers consisted of 6 females and 12 males between the ages of 20 and 36, with mainly white skin color.

As the UAS required a special ascent permit, the study had to be conducted in an open field that was available for FSD test flights. Since there was no other infrastructure, especially for charging the UAS’s batteries and connecting electronic hardware, the study required a significant amount of organizational and personnel effort. Therefore, all recordings of the study were performed on a single day in winter.

The general setup of the recording is shown in Figure 6. The aim was to record the volunteers at a viewing distance of 5 m and at a horizontal angle of 45°. Since no live view was available during the recordings, a fixed position was measured for the volunteers in advance of the recordings, which the UAS permanently targeted during the flight. For each recording, the volunteers were asked to sit in the same exact position (Figure 6, (3)). Two consecutive recordings were captured for each subject. These recordings were referred to as Phases I and II. Recording time was limited to 30 s to better accommodate the drone’s limited flight time and the triage aspects, as rapid assessments were a priority. To simulate different HRs, the 18 volunteers were divided in three groups of 6 each. In the first group, resting HR conditions were considered. In the second and third groups, the aim was to simulate “abnormal” physiological conditions. To achieve higher than usual HRs during Phase I and eventually Phase II, group two was instructed to exercise heavily immediately before recording in order to simulate tachycardia. Group three was instructed to breathe as slowly as possible during Phase I to lower the HR gradually. The recordings were conducted during the daytime.

For validation purposes, the vital parameters of the volunteers were assessed with a patient monitor, Philips MP2 (Koninklijke Philips N.V., Amsterdam, The Netherlands). Only the pulse oximetry was recorded at a sampling rate of 100 HZ.

## 3. Results

### 3.1. Recording Conditions

The recordings were conducted during daylight between 11 a.m. and 4 p.m. During the recordings, the external weather conditions were harsh. The average wind force on the day of the recording was 6–7, corresponding to a wind speed of about 14 m/s. In addition, the weather was highly dynamic between sun, clouds, and light rain, which resulted in constantly changing lighting conditions. The outside temperature was about 7 °C.

### 3.2. Target Acquisition and Adaptive Exposure Time Adjustment

The ability of the UAS to maintain the target acquisition area and the subsequent adjustments of the exposure time was evaluated based on its ability to provide appropriate time series of more than 15 s (i.e., 225 samples) as input for the presented approach.

The gimbal could not always compensate for the UAS displacement caused by the gusty winds on the recording day, as the wind pushed the UAS up to 80 cm away from its designated position. Due to these uncompensated shifts, the most crucial area-of-interest, the face, was entirely or partially lost during the recording. The complete or partial loss of the face consequently affected the automatic adjustment of the exposure times, which used the pixel values of this ROI as input. This loss led to underexposed or overexposed images, making HR assessments impossible. It also resulted in the inability to generate a continuous time series of facial regions, which made it impossible to determine the HR as well. In total, 12 out of the 36 recordings could not be analyzed and were removed from the assessment. In six of the failed recordings, the time-series length was insufficient for HR extraction, and in the other six, the faces were overexposed. An overview is shown in Table 2.

Of the other 24 recordings, an accumulated time-series duration of well over 20 s was achieved in 9 sets. A total of 3 recordings were under 10 s, and the remaining were in between 10 s and 20 s. Regarding the face-related exposure settings, in 14 of the 24 successful recordings, a mean facial pixel value $\bar{v}$ within the target range of above 190 without signal clipping could be achieved. The remaining were split into 5 recordings that were underexposed ($\bar{v}$ < 190) and 5 recordings that were overexposed ($\bar{v}$ > 250).

### 3.3. Heart Rate Assessment

The overall performance of the proposed approach for all valid datasets is shown in Figure 7. The HR assessed by the UAS was compared against the HR-rate value provided by the reference pulsometer via a Bland Altman plot. The reference was calculated as the mean value of the HR values for the valid time points of the video recording. The combined value of the CHROM- and POS-assessed HRs produced an almost mean-value-free distribution of the determined values. Most values fell within the range of ±20 bpm and were in the 95% confidence interval of −28.5–27.5 bpm.

To investigate the influence of the system’s motion and exposure-time adjustments on the quality of the HR assessments, we split the 24 recordings into different subsets. Concerning system motion, we considered the total length of the time series extracted by the pipeline. The more frequently the face was lost due to the system motion, the shorter the total length. This way, the dataset was split into 11 recordings, representing high system motion when the length was shorter than 15 s (i.e., 225 samples) and 13 recordings of low system motion when the length was greater than 15 s. Regarding the exposure, the set was split into the 14 recordings where the facial color-channel values were within the desired range and the 10 recordings where they were not. The cross-relations for all color-mixing methods regarding the root-mean-squared errors (RMSEs) of all four sets, as well as the complete dataset, are shown in Table 3.

The RMSE values showed that the approach benefited most from a more stable recording and good exposure (low motion, inside exposure range), resulting in an RMSE of 11.4, which was comparable to the other methods. The effects of image stability and exposure were comparable in our recordings. Optimal illumination (high and low motion, inside exposure range), as compared to a less optimal environment (high and low motion, outside exposure range), led to an improvement of the RMSE by 4.8 bpm, whereas a stable recording (low motion, all exposure ranges), as compared to an unstable one (high motion, all exposure ranges), led to an improvement of 3.5 bpm. Furthermore, the respective worst-case results (high motion, all exposure ranges) with 16.0 bpm vs. (high and low motion, outside exposure range) 16.8 bpm, and the best-case results (low motion, all exposure ranges) with 12.5 bpm vs. (high and low motion, inside exposure range) 12 bpm, were very similar. Both indicated that the influence of the exposure and the significant system motion had an equal effect on the quality of the acquired HR assessments.

## 4. Discussion

Our research represented an initial exploration of the feasibility of acquiring vital parameters via drones under outdoor conditions. Furthermore, it evaluated the possibility of automatically assessing individual HRs in the context of MCI scenarios and using a specialized UAS. The data presented in the study were obtained through a feasibility investigation involving a small group of stationary and easily observable subjects. Therefore, the findings should be regarded as preliminary indications of the feasibility and serve only as a foundation for future research. To validate the findings and draw conclusive inferences, further studies with a larger sample of subjects representative of the population and a more extensive range of measured HRs are necessary, particularly under more realistic conditions.

Despite the challenging weather conditions during the study, we found that the developed tilt-wing UAS was capable of producing recordings appropriate for HR assessments. However, we also found that recording an individual’s face for a sufficient length of time during powerful wind gusts was not feasible. In an actual application, this would have to be detected automatically, so the recording time could be extended accordingly.

Losing the facial region also affected the adaptive exposure. In the 14 flawless working recordings, the face was continuously within the frames during the adjustment. For the other recordings, the algorithm either did not adjust because no facial ROIs were recognized or arbitrary regions were incorrectly recognized as facial ROIs. The latter was mainly responsible for the highly overexposed recordings. If more powerful hardware becomes available, more robust, deep neural-network-based algorithms for facial detection could be used [33]. The fusion of several sensors, e.g., by using an additional thermal-imaging camera, could also ensure that only actual facial ROIs are detected.

Our results showed, additionally, that the general assessment of HRs via drones was possible. Based on the RMSE with slight system motion and sufficient exposure, the values achieved in our study were comparable with the results of other rPPG studies, which had been performed under much more controlled conditions. The 11.4 bpm RMSE we achieved was similar to the results of Nowara et al., who had achieved an RMSE of 11.0 ± 3.8 bpm while driving under motion and using a similar approach [34]. However, the high-motion results showed much room for improvements in the proposed system. A more stable image, in which the face was consistently captured, and good illumination of the face could contribute equally to better results, lowering the RMSE of 19.3 bpm to 13.9 bpm and 12.7 bpm, respectively.

In contrast to a real-world scenario where higher HRs are caused by stress or medical conditions, the HRs after brief physical activity, such as those in our study, only remained stable for the limited duration of a few seconds before rapidly decreasing and returning to near-resting levels within one minute. This limited the evaluation of our method, as it relied on a stable HR over time and, thus, could result in less accurate detection of higher HRs. However, the data collected in our study, as displayed in Figure 7, did not demonstrate significantly higher deviations at higher HRs. One potential explanation for this could be that the valid recording time periods in these cases were well below 20 s, thereby reducing the impact of this effect.

The achieved average RMSE of 14.3 bpm showed that our approach could not provide the precise HR measurements recorded in a clinical setting. However, it was suitable for general HR estimations during MCIs. In such scenarios, the exact HR is not the focus, but rather, determining whether the HR falls above or below specific thresholds is critical. Additionally, a manual pulse assessment, commonly used in emergency services, can also deviate by several bpm from the actual HR, particularly in overly stressful situations such as MCIs. When considering the worst-case results of almost 20 bpm RMSE, the pulse detection was well above the acceptable margin-of-error in triage. To address this limitation, future research could focus on obtaining more accurate results and developing an error measure that evaluates the quality of the respective measurements and notates poor measurements as unreliable.

Using a 3-axis gimbal integrated into the UAS could solve the most significant issues impeding the reliability of the system. One of the main reasons for the loss of the face region during recording and related overexposure was that the 2-axis gimbal could not compensate for the central disturbance axis. In addition, our results suggested that using a 3-axis gimbal could also increase the accuracy of the acquisition, as both the exposure and the available recording lengths would be improved. The algorithms for automatic exposure-time setting and robust facial recognition are typically integrated into the hardware of most modern consumer cameras. However, to our knowledge, no commercially available product combines a 3-axis gimbal with this camera technology that can also provide the images in raw format for evaluation. This currently makes the use of specialized drones with their own recording hardware a requirement.

Current research in rPPG is focused on training machine-learning-based methods for HR assessments. Previous research has shown that with more available training data, machine-learning-based methods have inevitably outperformed conventional approaches, which has also been shown in face-mesh-generation networks. Furthermore, due to our module-based approach, each step could be replaced by a machine-learning-based method to improve performance successively in the future. Improvements in recording the rPPG signal could also allow other blood-flow-related measurements to be assessed, such as blood pressure [35], which is also relevant for emergency medical services.

However, using visible light cameras, in general, and remote rPPGs, in particular, included their inherent limitations. It was essential that a person’s face be recorded while unobscured, meaning that individuals not facing the drone could not be assessed at all. In addition, the facial ROIs should, if possible, not be covered. Although previous research has demonstrated that with facial hair and makeup, an rPPG measurement was still possible [36], it could be assumed that heavy occlusions by blood and dirt, which would be expected in MCI scenarios, could significantly impair the results of an HR assessment.

Additionally, the accuracy of HR detection using rPPGs was highly dependent on the amount of light reflected from the skin, which varies depending on the individual’s skin type. Individuals with higher melanin content in their skin, which absorbs a greater proportion of environmental light than those with lower melanin content, tend to have less accurate rPPG measurements [25,37]. This could result in people with darker skin tones being disadvantaged during technically assisted triage. For these reasons, using a visible light camera alone may not be sufficient for reliable HR assessment during MCIs. Instead, it would be necessary to incorporate additional sensors, such as radar, which could record vital parameters and the visibility of certain facial regions, independently of skin color. However, it should be noted that radar systems are generally more expensive and more susceptible to system movements than cameras.

Theoretically, our approach could also be used on consumer-grade drones. They have the advantage that they are already available for most emergency services and first-responder units. Furthermore, these drones have the necessary hardware to perform all the steps of our pipeline. However, the necessary data from the flight system sensors, the camera gimbal, and the raw camera data are typically unavailable to the end-users due to manufacturer restrictions, which limited our research to specialized flight systems.

Our special-purpose flight system had the advantage that its tilt-wing design and autonomous flight enabled it to be sent out simultaneously with the first-responders since, as compared to conventional drones, it could travel long distances. Thereby, it arrived on the scene before the emergency services. Along with the improved recording hardware and the possibility of assessing additional parameters relevant to triage, this could enable it to perform an automated triage in advance for emergency services.

In order to apply the presented approach in real-world applications, it is necessary to conduct further research on several factors that were intentionally disregarded in our study but are expected to affect the reliability during real MCIs. These include, but are not limited to, individuals in different poses, moving individuals, partially concealed individuals, covered faces, and individuals not facing the camera. Additionally, distinguishing between emergency responders and injured individuals is another critical factor that needs to be explored in an actual application.

## 5. Conclusions

This paper presented the first detailed study of the usage of rPPGs for HR assessments using a UAS in the context of MCI scenarios. We presented a detailed approach, incorporating both hardware and software concepts, for the most robust assessments possible. First, we detailed the hardware settings and necessary sensor information to design an HR assessment pipeline. Second, we showed that with this information, an assessment was possible within the scope of an estimation, despite the harsh environmental influences. The system could not evaluate one-third of the available records because the movement compensation required due to harsh weather; however, with moderate system motion, results were achieved that could compete with other approaches under controlled conditions. This marks the first step towards an automated triage system during MCIs. However, further research is needed to explore the impact of movement, body position, and the concealment of facial regions and to develop a method for identifying unreliable measurements.

## Figures and Tables

**Figure 1 bioengineering-10-00336-f001:**
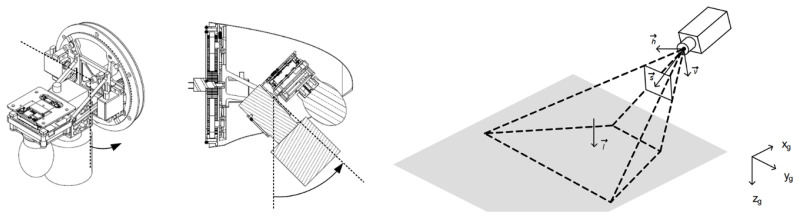
(**Left**) Technical drawing of the available gimbals in perspective and side views. Shown is the suspension of the sensors and the movable axes. (**Right**) The view frustum generated by the camera gimbals and the view vectors.

**Figure 2 bioengineering-10-00336-f002:**
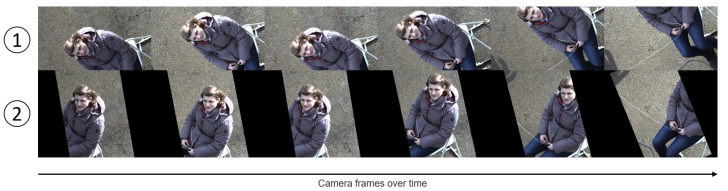
Time series of recorded camera frames before (1) and after (2) image stabilization.

**Figure 3 bioengineering-10-00336-f003:**
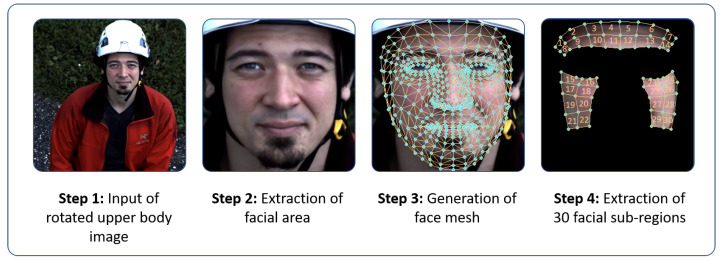
Individual steps for facial ROI selection and tracking.

**Figure 4 bioengineering-10-00336-f004:**
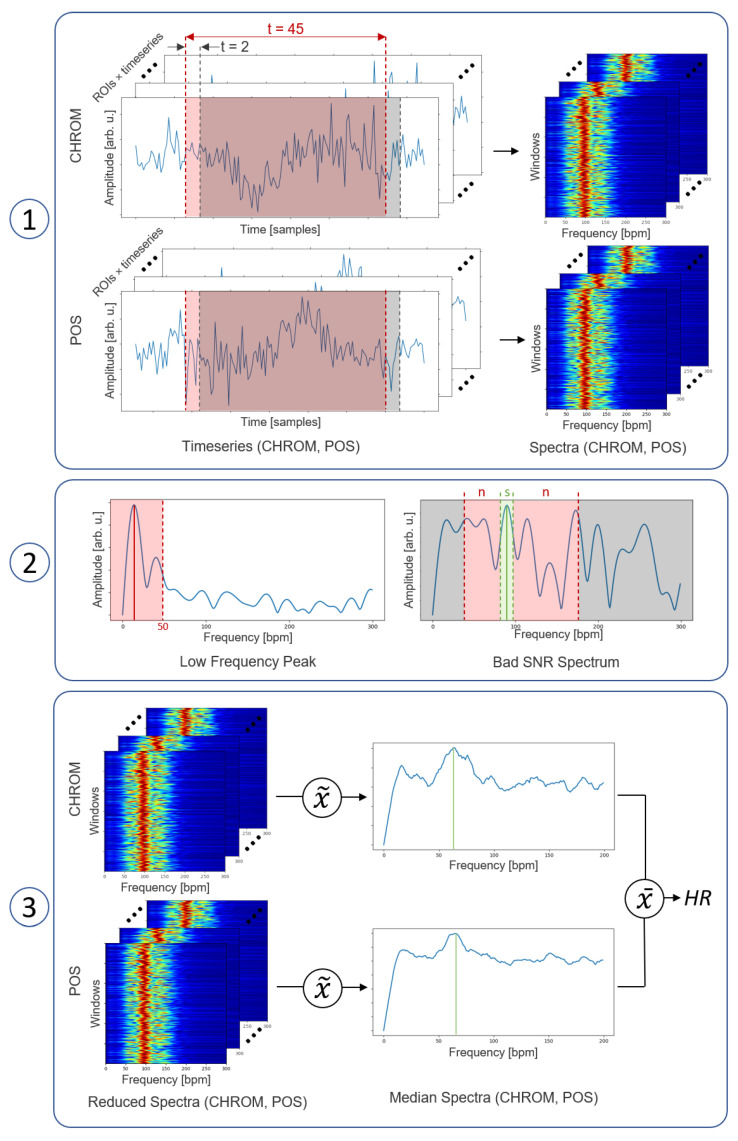
Overview of the HR estimation process—Windowing, time-series creation, and spectra calculation (1). Rejection of outlier spectra with low-frequency components or insufficient SNR (2). Final HR assessment by median spectra creation (3).

**Figure 5 bioengineering-10-00336-f005:**
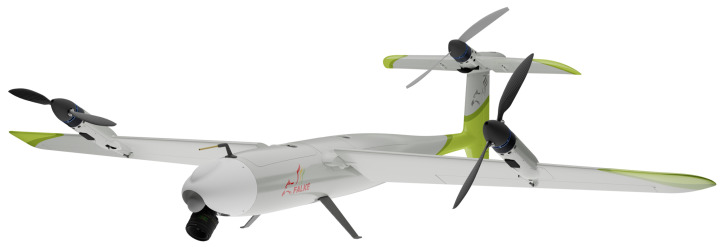
3D rendering of the “Neo” tilt-wing UAS by flyXdrive used in the study.

**Figure 6 bioengineering-10-00336-f006:**
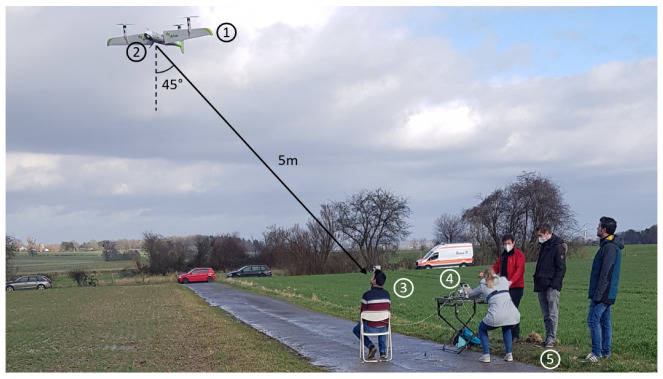
Experimental setup: UAS (1) equipped with gimbal and camera (2) was hovering within a 5 m viewing distance and target engagement angle of 45° of a volunteer subject at a fixed position (3). The reference pleth signal was recorded by a patient monitor and stored on a computer (4). The rest of the respective subject group was preparing for their recordings (5).

**Figure 7 bioengineering-10-00336-f007:**
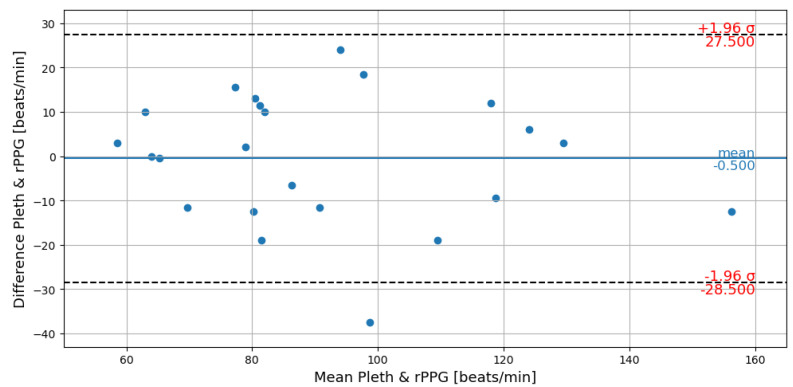
Bland–Altman plot of reference Pleth and rPPG.

**Table 1 bioengineering-10-00336-t001:** List of known 15 noise sources and their individual components that were used in the de-noising process.

landmarks (4–6) (x,y)	center-of-mass (x,y)	number of pixels
$\frac{d}{dt}$landmarks (4–6) (x,y)	$\frac{d}{dt}$center-of-mass (x,y)	height above ground
longitude	latitude	velocity (u,v,w)
nostrils (center, right)	eyes (center, right)	$\vec{h}$ (x,y,z)
$\vec{v}$ (x,y,z)	$\frac{d}{dt}\vec{h}$ (x,y,z)	$\frac{d}{dt}\vec{v}$ (x,y,z)

**Table 2 bioengineering-10-00336-t002:** Overview of valid frames and achieved mean facial exposure for each subject and recording phases.

		Phase I			Phase II	
**Subject**	**Valid Frames**	**Mean Exposure**	**Observation**	**Valid Frames**	**Mean Exposure**	**Observation**
S01	423	208		61	215	
S02	306	185		-	-	face out of area
S03	212	253		350	252	
S04	-	-	face out of area	-	-	face out of area
S05	-	-	over exposure	163	229	
S06	-	-	over exposure	-	-	over exposure
S07	219	212		260	214	
S08	327	244		-	-	face out of area
S09	49	230		390	114	
S10	390	147		390	230	
S11	213	230		394	230	
S12	218	180		280	232	
S13	288	163		208	252	
S14	-	-	over exposure	-	-	face out of area
S15	279	253		220	252	
S16	170	225		68	253	
S17	365	228		-	-	face out of area
S18	-	-	over exposure	-	-	over exposure

**Table 3 bioengineering-10-00336-t003:** RMSE bpm for the different subsets according to the recording conditions.

	Recordings Outside Exposure Range		Recordings Inside Exposure Range		All Exposure Ranges	
	Number of Recordings	RMSE (bpm)	Number of Recordings	RMSE (bpm)	Number of Recordings	RMSE (bpm)
**High Motion Recordings**	5	19.3	6	12.7	11	16.0
**Low Motion Recordings**	5	13.9	7	11.4	12	12.5
**High and Low Motion Recordings**	10	16.8	13	12.0	23	14.3

## Data Availability

Not applicable.

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
