# Peer review of "For Heart Rate Assessments from Drone Footage in Disaster Scenarios"

_bioengineering, 2023, doi:10.3390/bioengineering10030336_

Round 1

Reviewer 1 Report (Previous Reviewer 4)

The study used drone to detect the heart rate of the victims in a mass casualty incident (MCI). The difference of the study from others is the use of drone so that there are additional work to overcome the interference to the image capture and analysis. However, there is a major flaw in the applicability of the system in the proposed scenario of MCI. In the MCI, the faces of the victims are less likely clean and facing the camera, and could be covered with blood or dirt. Those factors were not discussed in the paper and can fail the technology completely.

Minor issues:

1. Line 44: What is "i" in iPPG?

2. Line 148: What is "the related"?

3. Line 386: Change "often" the "frequently"

Author Response

Thank you for your insightful comments on our manuscript. 

We acknowledge the considerations you raised regarding the visibility of victims' faces and potential issues caused by blood or dirt in the context of MCI, which may affect the applicability of our proposed system. Therefore, in addition to the discussed limitations of rPPG we have further addressed these limitations in our discussion section and highlighted the additional requirements that need to be fulfilled before the approach can be considered for actual use. 

Regarding the minor issues you raised, we made the following changes:

1) There was a missing "Image" referring to image PPG (iPPG), which was added.

2) This refers to the vector h, which is left facing related to the view axis.

3) Was changed to "frequently"

Thank you again for your valuable feedback, which will help us improve the quality of our manuscript.

Reviewer 2 Report (Previous Reviewer 2)

The authors has successfully addressed all the comments.

Author Response

Thank you for your positive feedback. We appreciate your time and effort in reviewing our manuscript and providing constructive comments that helped us improve the quality of our work. We are glad that our revisions have satisfactorily addressed your concerns and suggestions. If you have any further comments or questions, please do not hesitate to contact us.

Reviewer 3 Report (New Reviewer)

This revised paper has improved and replied to reviewer's comments by summarizing the research aims, objectives, and engineering contributions.

Author Response

Thank you for your positive feedback. We appreciate your time and effort in reviewing our manuscript and providing constructive comments that helped us improve the quality of our work. We are glad that our revisions have satisfactorily addressed your concerns and suggestions. If you have any further comments or questions, please do not hesitate to contact us.

This manuscript is a resubmission of an earlier submission. The following is a list of the peer review reports and author responses from that submission.

Round 1

Reviewer 1 Report

we do believe this is a good engineering work. However, for the research paper, we can’t not clearly find a scientific problem and key points, let alone the innovation.

Advice: Cause the authors are trying to realize the HR estimation using UVA-based camera, so what’s the most challenging problem in this specific situation need to be find out and address. Otherwise, it's just a great combination of  some good technologies.

Author Response

Dear reviewers.

We appreciate your feedback on our manuscript and agree that identifying a scientific problem and key points is crucial for a research paper. We want to clarify that the primary objective of our study was to investigate the feasibility of using UVA-based heart rate estimation in real-world scenarios, specifically in the context of mass casualty incidents. The main challenge in this situation is the ability to accurately estimate heart rate in adverse conditions, such as changing light and heavy system motion. As there are yet to be comparable approaches, this still poses a scientific problem instead of a mere engineering one. We have revised the manuscript to highlight the scientific problem and key points better, as well as the potential innovation of our approach, by adding our research aims, challenges, and objectives to the introduction.

Reviewer 2 Report

Authors design Towards Robust Heart Rate Assessment from Drone Footage in Disaster Scenarios. The idea and contributions of paper are good. But paper needs some minor modification.

[1]    Introduction need to be more extensive and missing some description.

[2]   Research Problem must be added with heading under section Introduction.

[3]   Aims and Objectives of the study must be separately included in Introduction Section.

[4]   Discussion needs to be some more extensive.

[5]    There is no mention of limitations and future work in the study.

[6]   Include some more latest references.

Author Response

Dear Reviewer,
We appreciate your feedback on our manuscript and
thank you for taking the time to provide detailed and constructive comments. We have taken your suggestions into consideration and have made the necessary revisions to improve the quality of our work.

In response to your comments, we have:

[1] Extended and restructured the introduction to include a more extensive description of our research aims, challenges, and objectives, as well as a more detailed explanation of our contributions.

[2] Added a section on the research problem under the introduction, in coherence with point 1.

[3] Addressed the aims and objectives in the introduction section. However, we did not add separate subheadings as they would not fit the usual style of the journal.

[4] Extended the discussion by including study limitations, applications during mass casualty incidents, and future research. Additionally, the limitations of the RPPG approach related to obscured skin and skin color were discussed.

[5] Included a discussion on limitations and future work, in coherence with point 4.

[6] Included two additional recent references to reinforce statements in the discussion.

We hope that these revisions will meet your expectations and that our manuscript will be suitable for publication. We look forward to your further comments.

Reviewer 3 Report

Words of opinion:

The proposed manuscript describes a very interesting project with a very ambitious technological goal. The authors proposed to develop and test a technology to be mounted on drones for rapid assessment of humans during mass casualty incidents. This is a valid application of rPPG but a very ambitious one. The paper describes clearly the important factors affecting the feasibility of the recordings and the constraints of both the technology and the situation. Ultimately, the evaluation of the technology is done for the most difficult situation, namely, windy days with a drone located far from the participants. It is remarkable that the authors did select such a situation and perhaps are a bit naive to expect that the technology would provide good results based on the current state of the art with rPPG. 

One notes that drones may be very helpful in bringing monitoring technology to individuals who are isolated (with or without the context of MCI), and there is no reason why the drome could not land close to the individuals to provide measurements with more stability (addressing one of the most constraint aspects of rPPG). Furthermore, by getting closer to the person, the drone could also provide a light source during the rPPG to ensure that facial illumination is sufficient. An aspect that has not been thoroughly discussed in the manuscript (hour of the day, level of ambient illumination). If the technical description of the paper is satisfactory, one notes that the conclusion is not aligned with the results of the study. The level of error in HR measurement is not meeting the medical-grade expectations hence there is no evidence that the proposed technology could actually work for the proposed application. This statement is also reinforced by the fact that the range of measured HR in the available set of data is very weak, especially for bradycardic rhythms. Hence the conclusion should be adjusted and aligned with the presented results. Finally, a complete spectrum of skin complexions should be included in any research work on rPPG. Melanin is one of the barriers significantly diminishing the rPPG signal strength,  any technologies which would discriminate between human skin tone in case of patients triage has no future. This aspect cannot just be neglected. 

On a final note, the manuscript reports an interesting ambitious development that is worth publishing. Yet, the authors should consider emphasizing that the method has strong limitations with a low collection rate and an HR accuracy that is not medical grade and adds a section about study limitations.

Specific comments:

Title: you might consider removing the adjective ”robust” from the title. Achieving HR measurement with drone footage sounds good enough.

Line 313: more information is needed to understand how elevated HR was performed. In healthy individuals, the recovery time to resting HR can take only a few seconds (20 sec) which means that the pulse is not constant during this period and the use of spectral analysis to extract HR fails (by spectral requirement definitions). The results should report the rate of failure during resting vs. in recovery after intense exercise.

Author Response

We would like to thank you for your thoughtful and detailed feedback on our manuscript. We have carefully considered all of the comments and have made the necessary revisions to improve the quality of our work.

The suggestion of utilizing a drone that lands close to an individual to obtain more stable heart rate measurements and improved facial illumination is an interesting concept. However, several practical limitations need to be considered. The drone would need to be small and lightweight to avoid posing additional hazards to the individual. Additionally, such a design would likely be limited to a multicopter design, which is not capable of covering large distances as our presented design. Hence it has to be transported to the incident site first. Furthermore, additional time is required for departure and landing at each location, which may not be feasible in the context of MCIs where speed and efficiency are crucial. Therefore, we have not included this approach in the revised manuscript.

Regarding the aspect of ambient illumination we added the time of the experiment to the description of the weather and lighting conditions in Section 3.1.

We also have clarified in the discussion that our results should not be considered representative or sufficient for medical-grade monitoring. However, we have also provided additional context on the typical heart rate measurement method used in MCIs, and have argued that our achieved average RMSE is suitable for triage purposes, thereby warranting further research. Additionally, we have extended the limitations of our study, including the worst-case results that were not satisfactory and require further investigation.

Additionally, the implications of the study's design regarding measuring higher heart rates have been included in the discussion section of the revised manuscript.

Regarding your concern about the limitations of our approach concerning skin color, we totally agree and therefore have added a comprehensive section to the discussion addressing the limitations of the rPPG approach for individuals with darker skin tones.

Lastly, the adjective "robust" was removed from the title as this would be more appropriate for future approaches.

We would like to express our gratitude for your valuable feedback and insights on our manuscript and are looking forward to your comments.

Reviewer 4 Report

The study tried to detect the heart rate remotely using rPPG in the disaster scenarios. The challenge addressed in the paper is the stability of the drone during the measurement. However, another factor affecting the outcome was not discussed, i.e. the faces of those victims in a mass casualty incidence are not clean and could be covered with dirt and blood. The authors should comment the usability of the technology in those cases.

Minor issues:

1. Line 24: change "mere" to "a few".

2. Line 33: Are there any word(s) missing after "considerably"?

3. Line 171: is "and ROI" "in ROI"?

4. Equation 3: Are the subscripts for y, p and e indexes of the frames? Please also specify the indexes of m and x.

Author Response

We would like to thank the you for your thoughtful and detailed feedback on our manuscript. We have carefully considered all of the comments and have made the necessary revisions to improve the quality of our work.

We agree that the presence of dirt and blood on the face of victims in a mass casualty incident could potentially impact the accuracy of rPPG measurements. We have added a section to discussion on this limitation in the revised manuscript.

Regarding your minor issues:

  1. “Mere” was changed to “few”
  2. Was changed to “considerably well”
  3. “and ROI” was changed to “and each ROI”
  4. We added anadditional description on y, and mx.

We would like to express our gratitude for your valuable feedback and insights on our manuscript, and looking forward to your comments.